# Effects of Data Geometry in Early Deep Learning

**Saket Tiwari**
Department of Computer Science
Brown University
Providence, RI 02906
`saket_tiwari@brown.edu`

**George Konidaris**
Department of Computer Science
Brown University
Providence, RI 02906

## Abstract

Deep neural networks can approximate functions on different types of data, from images to graphs, with varied underlying structure. This underlying structure can be viewed as the geometry of the data manifold. By extending recent advances in the theoretical understanding of neural networks, we study how a randomly initialized neural network with piece-wise linear activation splits the data manifold into *regions* where the neural network behaves as a linear function. We derive bounds on the density of boundary of linear regions and the distance to these boundaries on the data manifold. This leads to insights into the expressivity of randomly initialized deep neural networks on non-Euclidean data sets. We empirically corroborate our theoretical results using a toy supervised learning problem. Our experiments demonstrate that number of linear regions varies across manifolds and the results hold with changing neural network architectures. We further demonstrate how the complexity of linear regions is different on the low dimensional manifold of images as compared to the Euclidean space, using the MetFaces dataset.

## 1 Introduction

The capacity of Deep Neural Networks (DNNs) to approximate arbitrary functions given sufficient training data in the supervised learning setting is well known [Cybenko, 1989, Hornik et al., 1989, Anthony and Bartlett, 1999]. Several different theoretical approaches have emerged that study the effectiveness and pitfalls of deep learning. These studies vary in their treatment of neural networks and the aspects they study range from convergence [Allen-Zhu et al., 2019, Goodfellow and Vinyals, 2015], generalization [Kawaguchi et al., 2017, Zhang et al., 2017, Jacot et al., 2018, Sagun et al., 2018], function complexity [Montúfar et al., 2014, Mhaskar and Poggio, 2016], adversarial attacks [Szegedy et al., 2014, Goodfellow et al., 2015] to representation capacity [Arpit et al., 2017]. Some recent theories have also been shown to closely match empirical observations [Poole et al., 2016, Hanin and Rolnick, 2019b, Kunin et al., 2020].

One approach to studying DNNs is to examine how the underlying structure, or geometry, of the data interacts with learning dynamics. The manifold hypothesis states that high-dimensional real world data typically lies on a low dimensional manifold [Tenenbaum, 1997, Carlsson et al., 2007, Fefferman et al., 2013]. Empirical studies have shown that DNNs are highly effective in deciphering this underlying structure by learning intermediate latent representations [Poole et al., 2016]. The ability of DNNs to "flatten" complex data manifolds, using composition of seemingly simple piece-wise linear functions, appears to be unique [Brahma et al., 2016, Hauser and Ray, 2017].

DNNs with piece-wise linear activations, such as ReLU [Nair and Hinton, 2010], divide the input space into linear regions, wherein the DNN behaves as a linear function [Montúfar et al., 2014]. The density of these linear regions serves as a proxy for the DNN's ability to interpolate a complex data landscape and has been the subject of detailed studies [Montúfar et al., 2014, Telgarsky, 2015, Serra

et al., 2018, Raghu et al., 2017]. The work by Hanin and Rolnick [2019a] on this topic stands out because they derive bounds on the average number of linear regions and verify the tightness of these bounds empirically for deep ReLU networks, instead of larger bounds that rarely materialize. Hanin and Rolnick [2019a] conjecture that the number of linear regions correlates to the expressive power of randomly initialized DNNs with piece-wise linear activations. However, they assume that the data is uniformly sampled from the Euclidean space $\mathbb{R}^d$, for some $d$. By combining the manifold hypothesis with insights from Hanin and Rolnick [2019a], we are able to go further in estimating the number of linear regions and the average distance from *linear boundaries*. We derive bounds on how the geometry of the data manifold affects the aforementioned quantities.

To corroborate our theoretical bounds with empirical results, we design a toy problem where the input data is sampled from two distinct manifolds that can be represented in a closed form. We count the exact number of linear regions and the average distance to the boundaries of linear regions on these two manifolds that a neural network divides the two manifolds into. We demonstrate how the number of linear regions and average distance varies for these two distinct manifolds. These results show that the number of linear regions on the manifold do not grow exponentially with the dimension of input data. Our experiments do not provide estimates for theoretical constants, as in most deep learning theory, but demonstrate that the number of linear regions change as a consequence of these constants. We also study linear regions of deep ReLU networks for high dimensional data that lies on a low dimensional manifold with unknown structure and how the number of linear regions vary on and off this manifold, which is a more realistic setting. To achieve this we present experiments performed on the manifold of natural face images. We sample data from the image manifold using a generative adversarial network (GAN) [Goodfellow et al., 2014] trained on the curated images of paintings. Specifically, we generate images using the pre-trained StyleGAN [Karras et al., 2019, 2020b] trained on the curated MetFaces dataset [Karras et al., 2020a]. We generate *curves* on the image manifold of faces, using StyleGAN, and report how the density of linear regions varies on and off the manifold. These results shed new light on the geometry of deep learning over structured data sets by taking a data intrinsic approach to understanding the expressive power of DNNs.

## 2   Preliminaries and Background

Our goal is to understand how the underlying structure of real world data matters for deep learning. We first provide the mathematical background required to model this underlying structure as the geometry of data. We then provide a summary of previous work on understanding the approximation capacity of deep ReLU networks via the complexity of linear regions. For the details on how our work fits into one of the two main approaches within the theory of DNNs, from the expressive power perspective or from the learning dynamics perspective, we refer the reader to Appendix C.

### 2.1   Data Manifold and Definitions

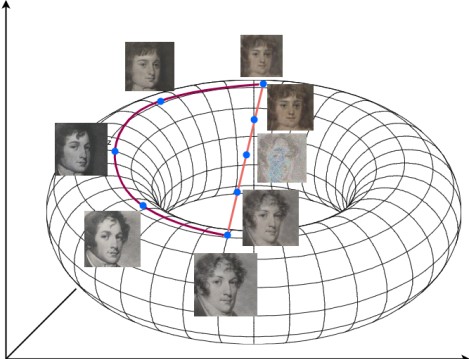

Figure 1: A 2D surface, here represented by a 2-torus, is embedded in a larger input space, $\mathbb{R}^3$. Suppose each point corresponds to an image of a face on this 2-torus. We can chart two curves: one straight line cutting across the 3D space and another curve that stays on the torus. Images corresponding to the points on the torus will have a smoother variation in style and shape whereas there will be images corresponding to points on the straight line that are not faces.

We use the example of the MetFaces dataset [Karras et al., 2020a] to illustrate how data lies on a low dimensional manifold. The images in the dataset are $1028 \times 1028 \times 3$ dimensional. By contrast, the number of *realistic* dimensions along which they vary are limited, e.g. painting style, artist, size and shape of the nose, jaw and eyes, background, clothing style; in fact, very few $1028 \times 1028 \times 3$ dimensional images correspond to realistic faces. We illustrate how this affects the possible variations in the data in Figure 1. A manifold formalises the notion of limited variations in high dimensional data. One can imagine that there exists an unknown function $f : X \to Y$ from a low dimensional space of variations, to a high dimensional space of the actual data points. Such a function $f : X \to Y$, from one open subset $X \subset \mathbb{R}^m$, to another open subset $Y \subset R^k$, is a *diffeomorphism* if $f$ is bijective, and both $f$ and $f^{-1}$ are differentiable (or smooth). Therefore, a manifold is defined as follows.

**Definition 2.1.** *Let $k, m \in \mathbb{N}_0$. A subset $M \subset \mathbb{R}^k$ is called a smooth $m$-dimensional submanifold of $\mathbb{R}^k$ (or $m$-manifold in $\mathbb{R}^k$) iff every point $x \in M$ has an open neighborhood $U \subset \mathbb{R}^k$ such that $U \cap M$ is diffeomorphic to an open subset $\Omega \subset \mathbb{R}^m$. A diffeomorphism (i.e. differentiable mapping),*

$$f : U \cap M \to \Omega$$

*is called a coordinate chart of M and the inverse,*

$$h := f^{-1} : \Omega \to U \cap M$$

*is called a smooth parametrization of $U \cap M$.*

For the MetFaces dataset example, suppose there are 10 dimensions along which the images vary. Further assume that each variation can take a value continuously in some interval of $\mathbb{R}$. Then the smooth parametrization would map $f : \Omega \cap \mathbb{R}^{10} \to M \cap \mathbb{R}^{1028 \times 1028 \times 3}$. This parametrization and its inverse are unknown in general and computationally very difficult to estimate in practice.

There are similarities in how geometric elements are defined for manifolds and Euclidean spaces. A smooth curve, on a manifold $M$, $\gamma : I \to M$ is defined from an interval $I$ to the manifold $M$ as a function that is differentiable for all $t \in I$, just as for Euclidean spaces. The shortest such curve between two points on a manifold is no longer a straight line, but is instead a *geodesic*. One recurring geometric element, which is unique to manifolds and stems from the definition of smooth curves, is that of a *tangent space*, defined as follows.

**Definition 2.2.** *Let $M$ be an $m$-manifold in $\mathbb{R}^k$ and $x \in M$ be a fixed point. A vector $v \in \mathbb{R}^k$ is called a tangent vector of $M$ at $x$ if there exists a smooth curve $\gamma : I \to M$ such that $\gamma(0) = x, \dot{\gamma}(0) = v$ where $\dot{\gamma}(t)$ is the derivative of $\gamma$ at $t$. The set*

$$T_x M := \{\dot{\gamma}(0) | \gamma : \mathbb{R} \to M \text{ is smooth} \gamma(0) = x\}$$

*of tangent vectors of $M$ at $x$ is called the tangent space of $M$ at $x$.*

In simpler terms, the plane tangent to the manifold $M$ at point $x$ is called the tangent space and denoted by by $T_x M$. Consider the upper half of a 2-sphere, $S^2 \subset \mathbb{R}^3$, which is a 2-manifold in $\mathbb{R}^3$. The tangent space at a fixed point $x \in S^2$ is the 2D plane perpendicular to the vector $x$ and tangential to the surface of the sphere that contains the point $x$. For additional background on manifolds we refer the reader to Appendix B.

## 2.2 Linear Regions of Deep ReLU Networks

The higher the density of these linear regions the more complex a function a DNN can approximate. For example, a $\sin$ curve in the range $[0, 2\pi]$ is better approximated by 4 piece-wise linear regions as opposed to 2. To clarify this further, with the 4 "optimal" linear regions $[0, \pi/2), [\pi/2, \pi), [\pi, 3\pi/2)$, and $[3\pi/2, 2\pi]$ a function could approximate the $\sin$ curve better than any 2 linear regions. In other words, higher density of linear regions allows a DNN to approximate the variation in the curve better. We define the notion of boundary of a linear regions in this section and provide an overview of previous results.

We consider a neural network, $F$, which is a composition of activation functions. Inputs at each layer are multiplied by a matrix, referred to as the weight matrix, with an additional bias vector that is added to this product. We limit our study to ReLU activation function [Nair and Hinton, 2010], which is piece-wise linear and one of the most popular activation functions being applied to various learning tasks on different types of data like text, images, signals etc. We further consider DNNs that map inputs, of dimension $n_{\text{in}}$, to scalar values. Therefore, $F : \mathbb{R}^{n_{\text{in}}} \to \mathbb{R}$ is defined as,

$$F(x) = W_L \sigma(B_{L-1} + W_{L-1}\sigma(...\sigma(B_1 + W_1 x))), \tag{1}$$

where $W_l \in \mathbb{M}^{n_l \times n_{l-1}}$ is the weight matrix for the $l^{\text{th}}$ hidden layer, $n_l$ is the number of neurons in the $l^{\text{th}}$ hidden layer, $B_l \in \mathbb{R}^{n_l}$ is the vector of biases for the $l^{\text{th}}$ hidden layer, $n_0 = n_{\text{in}}$ and $\sigma : \mathbb{R} \to \mathbb{R}$ is the activation function. For a neuron $z$ in the $l^{\text{th}}$ layer we denote the *pre-activation* of this neuron, for given input $x \in \mathbb{R}^{n_{\text{in}}}$, as $z_l(x)$. For a neuron $z$ in the layer $l$ we have

$$z(x) = W_{l-1,z}\sigma(...\sigma(B_1 + W_1 x)), \tag{2}$$

for $l > 1$ (for the base case $l = 1$ we have $z(x) = W_{1,z}x$) where $W_{l-1,z}$ is the row of weights, in the weight matrix of the $l^{\text{th}}$ layer, $W_l$, corresponding to the neuron $z$. We use $W_z$ to denote the weight vector for brevity, omitting the layer index $l$ in the subscript. We also use $b_z$ to denote the bias term for the neuron $z$.

Neural networks with piece-wise linear activations are piece-wise linear on the input space [Montúfar et al., 2014]. Suppose for some fixed $y \in \mathbb{R}^{n_{\text{in}}}$ as $x \to y$ if we have $z(x) \to -b_z$ then we observe a discontinuity in the gradient $\nabla_x \sigma(b_z + W_z z(x))$ at $y$. Intuitively, this is because $x$ is approaching the boundary of the linear region of the function defined by the output of $z$. Therefore, the boundary of linear regions, for a feed forward neural network $F$, is defined as:

$$\mathcal{B}_F = \{x | \nabla F(x) \text{ is not continuous at } x\}.$$

Hanin and Rolnick [2019a] argue that an important generalization for the approximation capacity of a neural network $F$ is the $(n_{\text{in}} - 1)-$dimensional volume density of linear regions defined as $\text{vol}_{n_{\text{in}}-1}(\mathcal{B}_F \cap K)/\text{vol}_{n_{\text{in}}}(K)$, for a bounded set $K \subset \mathbb{R}^{n_{\text{in}}}$. This quantity serves as a proxy for density of linear regions and therefore the expressive capacity of DNNs. Intuitively, higher density of linear boundaries means higher capacity of the DNN to approximate complex non-linear functions. The quantity is applied to lower bound the distance between a point $x \in K$ and the set $\mathcal{B}_F$, which is

$$\text{distance}(x, \mathcal{B}_F) = \min_{\text{neurons } z} |z(x) - b_z|/||\nabla z(x)||,$$

which measures the sensitivity over neurons at a given input. The above quantity measures how "far" the input is from flipping any neuron from inactive to active or vice-versa.

Informally, Hanin and Rolnick [2019a] provide two main results for a randomly initialized DNN $F$, with a reasonable initialisation. Firstly, they show that

$$\mathbb{E}\left[\frac{\text{vol}_{n_{\text{in}}-1}(\mathcal{B}_F \cap K)}{\text{vol}_{n_{\text{in}}}(K)}\right] \approx \#\{\text{ neurons}\},$$

meaning the density of linear regions is bound above and below by some constant times the number of neurons. Secondly, for $x \in [0,1]^{n_{\text{in}}}$,

$$\mathbb{E}\left[\text{distance}(x, \mathcal{B}_F)\right] \geq C\#\{\text{ neurons}\}^{-1},$$

where $C > 0$ depends on the distribution of biases and weights, in addition to other factors. In other words, the distance to the nearest boundary is bounded above and below by a constant times the inverse of the number of neurons. These results stand in contrast to earlier worst case bounds that are exponential in the number of neurons. Hanin and Rolnick [2019a] also verify these results empirically to note that the constants lie in the vicinity of 1 throughout training.

## 3 Linear Regions on the Data Manifold

One important assumption in the results presented by Hanin and Rolnick [2019a] is that the input, $x$, lies in a compact set $K \subset \mathbb{R}^{n_{\text{in}}}$ and that $\text{vol}_{n_{\text{in}}}(K)$ is greater than 0. Also, the theorem pertaining to the lower bound on average distance of $x$ to linear boundaries the input assumes the input uniformly distributed in $[0,1]^{n_{\text{in}}}$. As noted earlier, high-dimensional real world datasets, like images, lie on low dimensional manifolds, therefore both these assumptions are false in practice. This motivates us to study the case where the data lies on some $m-$dimensional submanifold of $\mathbb{R}^{n_{\text{in}}}$, i.e. $M \subset \mathbb{R}^{n_{\text{in}}}$ where $m \ll n_{\text{in}}$. We illustrate how this constraint effects the study of linear regions in Figure 2.

As introduced by Hanin and Rolnick [2019a], we denote the "$(n_{\text{in}} - k)-$dimensional piece" of $\mathcal{B}_F$ as $\mathcal{B}_{F,k}$. More precisely, $\mathcal{B}_{F,0} = \emptyset$ and $\mathcal{B}_{F,k}$ is recursively defined to be the set of points $x \in \mathcal{B}_F \setminus \{\mathcal{B}_{F,0} \cup ... \cup \mathcal{B}_{F,k-1}\}$ with the added condition that in a neighbourhood of $x$ the set $\mathcal{B}_{F,k}$

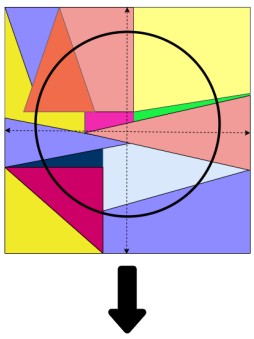

Figure 2: A circle is an example of a 1D manifold in a 2D Euclidean space. The effective number of linear regions on the manifold, the upper half of the circle, are the number of linear regions on the arc from $-\pi$ to $\pi$. In the diagram above, each color in the 2D space corresponds to a linear region. When the upper half of the circle is flattened into a 1D space we obtain a line. Each color on the line corresponds to a linear region of the 2D space.

coincides with hyperplane of dimension $n_{\text{in}} - k$. We provide a detailed and formal definition for $\mathcal{B}_{F,k}$ with intuition in Appendix E. In our setting, where the data lies on a manifold $M$, we define $\mathcal{B}'_{F,k}$ as $\mathcal{B}_{F,k} \cap M$, and note that $\dim(\mathcal{B}'_{F,k}) = m - k$ (Appendix E Proposition E.4). For example, the *transverse* intersection (see Definition E.3) of a plane in 3D with the 2D manifold $S^2$ is a 1D curve in $S^2$ and therefore has dimension 1. Therefore, $\mathcal{B}'_{F,k}$ is a submanifold of dimension $3 - 2 = 1$. This imposes the restriction $k \leq m$, for the intersection $\mathcal{B}_{F,k} \cap M$ to have a well defined volume.

We first note that the definition of the determinant of the Jacobian, for a collection of neurons $z_1, ..., z_k$, is different in the case when the data lies on a manifold $M$ as opposed to in a compact set of dimension $n_{\text{in}}$ in $\mathbb{R}^{n_{\text{in}}}$. Since the determinant of the Jacobian is the quantity we utilise in our proofs and theorems repeatedly we will use the term Jacobian to refer to it for succinctness. Intuitively, this follows from the Jacobian of a function being defined differently in the ambient space $\mathbb{R}^{n_{\text{in}}}$ as opposed to the manifold $M$. In case of the former it is the volume of the paralellepiped determined by the vectors corresponding to the directions with steepest ascent along each one of the $n_{\text{in}}$ axes. In case of the latter it is more complex and defined below. Let $\mathcal{H}^m$ be the $m-$dimensional Hausdorff measure (we refer the reader to the Appendix B for background on Hausdorff measure). The Jacobian of a function on manifold $M$, as defined by Krantz and Parks [2008] (Chapter 5), is as follows.

**Definition 3.1.** *The (determinant of) Jacobian of a function $H : M \to \mathbb{R}^k$, where $k \leq \dim(M) = m$, is defined as*

$$J^M_{k,H}(x) = \sup\left\{ \frac{\mathcal{H}^k(D_M H(P))}{\mathcal{H}^k(P)} \middle| P \text{ is a } k\text{-dimensional parallelepiped contained in } T_x M. \right\},$$

*where $D_M : T_x M \to \mathbb{R}^k$ is the differential map (see Appendix B) and we use $D_M H(P)$ to denote the mapping of the set $P$ in $T_x M$, which is a parallelepiped, to $\mathbb{R}^k$. The supremum is taken over all parallelepipeds $P$.*

We also say that neurons $z_1, ..., z_k$ are good at $x$ if there exists a path of neurons from $z$ to the output in the computational graph of $F$ so that each neuron is activated along the path. Our three main results that hold under the assumptions listed in Appendix A, each of which extend and improve upon the theoretical results by Hanin and Rolnick [2019a], are:

**Theorem 3.2.** *Given $F$ a feed-forward ReLU network with input dimension $n_{in}$, output dimension 1, and random weights and biases. Then for any bounded measurable submanifold $M \subset \mathbb{R}^{n_{in}}$ and any $k = 1, ...., m$ the average $(m-k)-$dimensional volume of $\mathcal{B}_{F,k}$ inside $M$,*

$$\mathbb{E}[vol_{m-k}(\mathcal{B}_{F,k} \cap M)] = \sum_{\text{distinct neurons } z_1,...,z_k \text{ in } F} \int_M \mathbb{E}[Y_{z_1,...,z_k}] dvol_m(x), \tag{3}$$

*where $Y_{z_1,...,z_k}$ is $J^M_{m,H_k}(x)\rho_{b_1,...,b_k}(z_1(x), ..., z_k(x))$, times the indicator function of the event that $z_j$ is good at $x$ for each $j = 1, ..., k$. Here the function $\rho_{b_{z_1},...,b_{z_k}}$ is the density of the joint distribution of the biases $b_{z_1}, ..., b_{z_k}$.*

This change in the formula, from Theorem 3.4 by Hanin and Rolnick [2019a], is a result of the fact that $z(x)$ has a different direction of steepest ascent when it is restricted to the data manifold $M$, for any $j$. The proof is presented in Appendix E. Formula 3 also makes explicit the fact that the data manifold has dimension $m \leq n_{\text{in}}$ and therefore the $m - k$-dimensional volume is a more representative measure of the linear boundaries. Equipped with Theorem 3.2, we provide a result for the density of boundary regions on manifold $M$.

**Theorem 3.3.** *For data sampled uniformly from a compact and measurable $m$ dimensional manifold $M$ we have the following result for all $k \leq m$:*

$$\frac{vol_{m-k}(\mathcal{B}_{F,k} \cap M)}{vol_m(M)} \leq \binom{\# \, neurons}{k} (2C_{grad}C_{bias}C_M)^k,$$

*where $C_{grad}$ depends on $||\nabla z(x)||$ and the DNN's architecture, $C_M$ depends on the geometry of $M$, and $C_{bias}$ on the distribution of biases $\rho_b$.*

The constant $C_M$ is the supremum over the matrix norm of projection matrices onto the tangent space, $T_x M$, at any point $x \in M$. For the Euclidean space $C_M$ is always equal to 1 and therefore the term does not appear in the work by Hanin and Rolnick [2019a], but we cannot say the same for our setting. We refer the reader to Appendix F for the proof, further details, and interpretation. Finally, under the added assumptions that the diameter of the manifold $M$ is finite and $M$ has polynomial volume growth we provide a lower bound on the average distance to the linear boundary for points on the manifold and how it depends on the geometry and dimensionality of the manifold.

**Theorem 3.4.** *For any point, $x$, chosen randomly from $M$, we have:*

$$\mathbb{E}[distance_M(x, \mathcal{B}_F \cap M)] \geq \frac{C_{M,\kappa}}{C_{grad}C_{bias}C_M \# neurons},$$

*where $C_{M,\kappa}$ depends on the scalar curvature, the input dimension and the dimensionality of the manifold $M$. The function $distance_M$ is the distance on the manifold $M$.*

This result gives us intuition on how the density of linear regions around a point depends on the geometry of the manifold. The constant $C_{M,\kappa}$ captures how volumes are distorted on the manifold $M$ as compared to the Euclidean space, for the exact definition we refer the reader to the proof in Appendix G. For a manifold which has higher volume of a unit ball, on average, in comparison to the Euclidean space the constant $C_{M,\kappa}$ is higher and lower when the volume of unit ball, on average, is lower than the volume of the Euclidean space. For background on curvature of manifolds and a proof sketch we refer the reader to the Appendices B and D, respectively. Note that the constant $C_M$ is the same as in Theorem 3.3. Another difference to note is that we derive a lower bound on the geodesic distance on the manifold $M$ and not the Euclidean distance in $\mathbb{R}^k$ as done by Hanin and Rolnick [2019a]. This distance better captures the distance between data points on a manifold while incorporating the underlying structure. In other words, this distance can be understood as how much a data point should change to reach a linear boundary while ensuring that all the individual points on the curve, tracing this change, are "valid" data points.

## 3.1 Intuition For Theoretical Results

One of the key ingredients of the proofs by Hanin and Rolnick [2019a] is the *co-area formula* [Krantz and Parks, 2008]. The co-area formula is applied to get a closed form representation of the $k-$dimensional volume of the region where any set of $k$ neurons, $z_1, z_2, ..., z_k$ is "good" in terms of the expectation over the Jacobian, in the Euclidean space. Instead of the co-area formula we use the *smooth co-area formula* [Krantz and Parks, 2008] to get a closed form representation of the $m - k-$dimensional volume of the region intersected with manifold, $M$, in terms of the Jacobian defined on a manifold (Definition 3.1). The key difference between the two formulas is that in the smooth co-area formula the Jacobian (of a function from the manifold $M$) is restricted to the tangent plane. While the determinant of the "vanilla" Jacobian measures the distortion of volume around a point in Euclidean space the determinant of the Jacobian defined as above (Definition 3.1) measures the distortion of volume on the manifold instead for the function with the same domain, the function that is 1 if the set of neurons are good and 0 otherwise.

The value of the Jacobian as defined in Definition 3.1 has the same volume as the projection of the parallelepiped defined by the gradients $\nabla z(x)$ onto the tangent space (see Proposition F.1 in

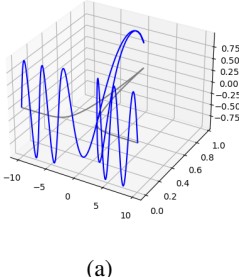 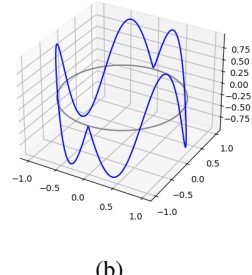

(a)                                          (b)

Figure 3: The tractrix (a) and circle (b) are plotted in grey and the target function is in blue. This is for illustration purposes and does not match the actual function or domains used in our experiments.

Appendix). This introduces the constant $C_M$, defined above. Essentially, the constant captures how the magnitude of the gradients, $\nabla z(x)$, are modified upon being projected to the tangent plane. Certain manifolds "shrink" vectors upon projection to the tangent plane more than others, on an average, which is a function of their geometry. We illustrate how two distinct manifolds "shrink" the gradients differently upon projection to the tangent plane as reflected in the number of linear regions on the manifolds (see Figure 11 in the appendix) for 1D manifolds. We provide intuition for the curvature of a manifold in Appendix B, due to space constraints, which is used in the lower bound for the average distance in Theorem 3.4. The constant $C_{M,\kappa}$ depends on the curvature as the supremum of a polynomial whose coefficients depend on the curvature, with order at most $n_{\text{in}}$ and at least $n_{\text{in}} - m$. Note that despite this dependence on the ambient dimension, there are other geometric constants in this polynomial (see Appendix G). Finally, we also provide a simple example as to how this constant varies with $n_{\text{in}}$ and $m$, for a simple and contrived example, in Appendix G.1.

# 4 Experiments

## 4.1 Linear Regions on a 1D Curve

To empirically corroborate our theoretical results, we calculate the number of linear regions and average distance to the linear boundary on 1D curves for regression tasks in two settings. The first is for 1D manifolds embedded in 2D and higher dimensions and the second is for the high-dimensional data using the MetFaces dataset. We use the same algorithm, for the toy problem and the high-dimensional dataset, to find linear regions on 1D curves. We calculate the exact number of linear regions for a 1D curve in the input space, $x : I \to \mathbb{R}^{n_{\text{in}}}$ where $I$ is an interval in real numbers, by finding the points where $z(x(t)) = b_z$ for every neuron $z$. The solutions thus obtained gives us the boundaries for neurons on the curve $x$. We obtain these solutions by using the programmatic activation of every neuron and using the sequential least squares programming (SLSQP) algorithm [Kraft, 1988] to solve for $|z(x(t)) - b_z| = 0$ for $t \in I$. In order to obtain the programmatic activation of a neuron we construct a Deep ReLU network as defined in Equation 2. We do so for all the neurons for a given DNN with fixed weights.

## 4.2 Supervised Learning on Toy Dataset

We define two similar regression tasks where the data is sampled from two different manifolds with different geometries. We parameterize the first task, a unit circle without its north and south poles, by $\psi_{\text{circle}} : (-\pi, \pi) \to \mathbb{R}^2$ where $\psi_{\text{circle}}(\theta) = (\cos \theta, \sin \theta)$ and $\theta$ is the angle made by the vector from the origin to the point with respect to the x-axis. We set the target function for regression task to be a periodic function in $\theta$. The target is defined as $z(\theta) = a \sin(\nu\theta)$ where $a$ is the amplitude and $\nu$ is the frequency (Figure 3). DNNs have difficulty learning periodic functions [Ziyin et al., 2020]. The motivation behind this is to present the DNN with a challenging task where it has to learn the underlying structure of the data. Moreover the DNN will have to split the circle into linear regions. For the second regression task, a tractrix is parametrized by $\psi_{\text{tractrix}} : \mathbb{R}^1 \to \mathbb{R}^2$ where $\psi_{\text{tractrix}}(y) = (y - \tanh y, \operatorname{sech} y)$ (see Figure 3). We assign a target function $z(t) = a \sin(\nu t)$. For the purposes of our study we restrict the domain of $\psi_{\text{tractrix}}$ to $(-3, 3)$. We choose $\nu$ so as to ensure

that the number of peaks and troughs, 6, in the periodic target function are the same for both the manifolds. This ensures that the domains of both the problems have length close to 6.28. Further experimental details are in Appendix H.

The results, averaged over 20 runs, are presented in Figures 4 and 5. We note that $C_M$ is smaller for Sphere (based on Figure 4) and the curvature is positive whilst $C_M$ is larger for tractrix and the curvature is negative. Both of these constants (curvature and $C_M$) contribute to the lower bound in Theorem 3.4. Similarly, we show results of number of linear regions divided by the number of neurons upon changing architectures, consequently the number of neurons, for the two manifolds in Figure 8, averaged over 30 runs. Note that this experiment observes the effect of $C_M \times C_{\text{grad}}$, since changing the architecture also changes $C_{\text{grad}}$ and the variation in $C_{\text{grad}}$ is quite low in magnitude as observed empirically by Hanin and Rolnick [2019a]. The empirical observations are consistent with our theoretical results. We observe that the number of linear regions starts off close to #neurons and remains close throughout the training process for both the manifolds. This supports our theoretical results (Theorem 3.3) that the constant $C_M$, which is distinct across the two manifolds, affects the number of linear regions throughout training. The tractrix has a higher value of $C_M$ and that is reflected in both Figures 4 and 5. Note that its relationship is inverse to the average distance to boundary region, as per Theorem 3.4, and it is reflected as training progresses in Figure 5. This is due to different "shrinking" of vectors upon being projected to the tangent space (Section 3.1).

### 4.3 Varying Input Dimensions

To empirically corroborate the results of Theorems 2 and 3 we vary the dimension $n_{\text{in}}$ while keeping $m$ constant. We achieve this by counting the number of linear regions and the average distance to boundary region on the 1D circle as we vary the input dimension in steps of 5. We draw samples of 1D circles in $\mathbb{R}^{n_{\text{in}}}$ by randomly choosing two perpendicular basis vectors. We then train a network with the same architecture as the previous section on the periodic target function $(a \sin(\nu\theta))$ as defined above. The results in Figure 6 shows that the quantities stay proportional to #$neurons$, and do not vary as $n_{\text{in}}$ is increased, as predicted by our theoretical results. Our empirical study asserts how the relevant upper and lower bounds, for the setting where data lies on a low-dimensional manifold, does not grow exponentially with $n_{\text{in}}$ for the density of linear regions in a compact set of $\mathbb{R}^{n_{\text{in}}}$ but instead depend on the intrinsic dimension. Further details are in Appendix H.

### 4.4 MetFaces: High Dimensional Dataset

Our goal with this experiment is to study how the density of linear regions varies across a low dimensional manifold and the input space. To discover latent low dimensional underlying structure of data we employ a GAN. Adversarial training of GANs can be effectively applied to learn a mapping from a low dimensional latent space to high dimensional data [Goodfellow et al., 2014]. The generator is a neural network that maps $g : \mathbb{R}^k \to \mathbb{R}^{n_{\text{in}}}$. We train a deep ReLU network on the MetFaces dataset with random labels (chosen from $0, 1$) with cross entropy loss. As noted by Zhang et al. [2017], training with random labels can lead to the DNN memorizing the entire dataset.

We compare the log density of number of linear regions on a curve on the manifold with a straight line off the manifold. We generate these curves using the data sampled by the StyleGAN by [Karras et al., 2020a]. Specifically, for each curve we sample a random pair of latent vectors: $z_1, z_2 \in \mathbb{R}^k$, this gives us the start and end point of the curve using the generator $g(z_1)$ and $g(z_2)$. We then generate 100 images to approximate a curve connecting the two images on the image manifold in a piece-wise manner. We do so by taking 100 points on the line connecting $z_1$ and $z_2$ in the latent space that are evenly spaced and generate an image from each one of them. Therefore, the $i^{\text{th}}$ image is generated as: $z_i' = g(((100 - i) \times z_1 + i \times z_2)/100)$, using the StyleGAN generator $g$. We qualitatively verify the images to ensure that they lie on the manifold of images of faces. The straight line, with two fixed points $g(z_1)$ and $g(z_2)$, is defined as $x(t) = (1 - t)g(z_1) + tg(z_2)$ with $t \in [0, 1]$. The approximated curve on the manifold is defined as $x'(t) = (1 - t)g(z_i') + tg(z_{i+1}')$ where $i = \texttt{floor}(100t)$. We then apply the method from Section 4.1 to obtain the number of linear regions on these curves.

The results are presented in Figure 9. This leads us to the key observation: the density of linear regions is significantly lower on the data manifold and devising methods to "concentrate" these linear regions on the manifold is a promising research direction. That could lead to increased expressivity for the same number of parameters. We provide further experimental details in Appendix I.

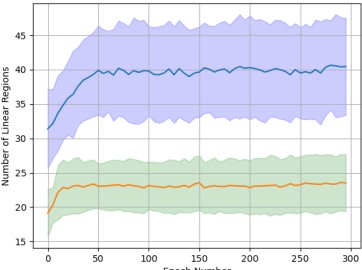

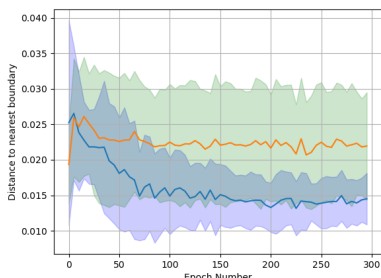

Figure 4: Graph of number of linear regions for tractrix (blue) and sphere (orange). The shaded regions represent one standard deviation. Note that the number of neurons is 26 and the number of linear regions are comparable to 26 but different for both the manifolds throughout training.

Figure 5: Graph of distance to linear regions for tractrix (blue) and sphere (orange). The distances are normalized by the maximum distance on the range, for both tractrix and sphere. The shaded regions represent one standard deviation.

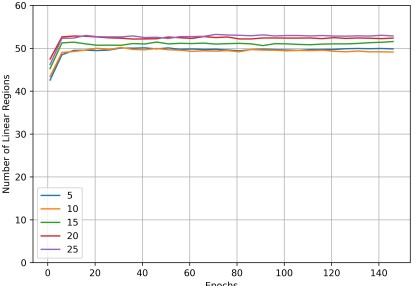

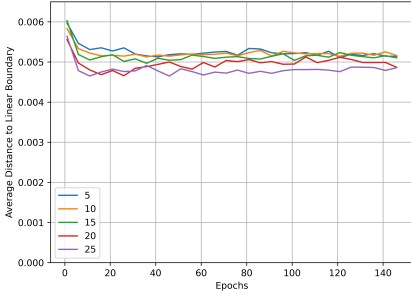

Figure 6: We observe that as the dimension $n_{\text{in}}$ is increased, while keeping the manifold dimension constant, the number of linear regions remains proportional to number of neurons (26).

Figure 7: We observe that as the dimension $n_{\text{in}}$ is increased, while keeping the manifold dimension constant, the average distance varies very little.

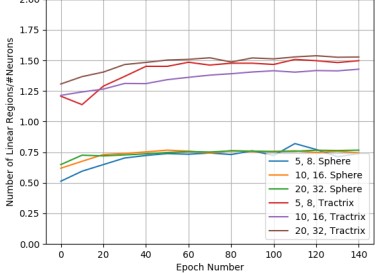

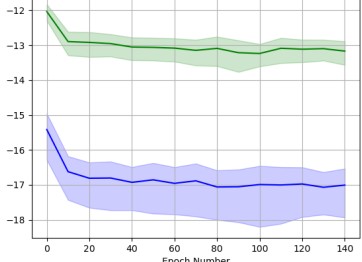

Figure 8: The effects of changing the architecture on the number of linear regions. We observe that the value of $C_M$ effects the number of linear regions proportionally. The number of hidden units for three layer networks are in the legend along with the data manifold.

Figure 9: We observe that the log density of number of linear regions is lower on the manifold (blue) as compared to off the manifold (green). This is for the MetFaces dataset.

## 5 Discussion and Conclusions

There is significant work in both supervised and unsupervised learning settings for non-Euclidean data [Bronstein et al., 2017]. Despite these empirical results most theoretical analysis is agnostic to data geometry, with a few prominent exceptions [Cloninger and Klock, 2020, Shaham et al., 2015, Schmidt-Hieber, 2019]. We incorporate the idea of data geometry into measuring the effective approximation capacity of DNNs, deriving average bounds on the density of boundary regions and distance from the boundary when the data is sampled from a low dimensional manifold. Our experimental results corroborate our theoretical results. We also present insights into expressivity of DNNs on low dimensional manfiolds for the case of high dimensional datasets. Estimating the geometry, dimensionality and curvature, of these image manifolds accurately is a problem that remains largely unsolved [Brehmer and Cranmer, 2020, Perraul-Joncas and Meila, 2013], which limits our inferences on high dimensional dataset to observations that guide future research. We note that proving a lower bound on the number of linear regions, as done by Hanin and Rolnick [2019a], for the manifold setting remains open. Our work opens up avenues for further research that combines model geometry and data geometry and can lead to empirical research geared towards developing DNN architectures for high dimensional datasets that lie on a low dimensional manifold.

## 6 Acknowledgements

This work was funded by L2M (DARPA Lifelong Learning Machines program under grant number FA8750-18-2-0117), the Penn MURI (ONR under the PERISCOPE MURI Contract N00014- 17-1-2699), and the ONR Swarm (the ONR under grant number N00014-21-1-2200). This research was conducted using computational resources and services at the Center for Computation and Visualization, Brown University.

We would like to thank Sam Lobel, Rafael Rodriguez Sanchez, and Akhil Bagaria for refining our work, multiple technical discussions, and their helpful feedback on the implementation details. We also thank Tejas Kotwal for assistance on deriving the mathematical details related to the 1D Tractrix and sources for various citations. We thank Professor Pedro Lopes de Almeida, Nihal Nayak, Cameron Allen and Aarushi Kalra for their valuable comments on writing and presentation of our work. We thank all the members of the Brown robotics lab for their guidance and support at various stages of our work. Finally, we are indebted to, and graciously thank, the numerous anonymous reviewers for their time and labor as their valuable feedback and thoughtful engagement have shaped and vastly refine our work.

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
