# OpenReview forum: "Effects of Data Geometry in Early Deep Learning"
_NeurIPS.cc/2022/Conference — NeurIPS 2022 Accept_

### Official Review · Reviewer_J3BE · 2022-07-10

**Rating:** 6
**Confidence:** 3
**Soundness:** 3 good
**Presentation:** 3 good
**Contribution:** 3 good

**Summary:**

The paper continues the work of Hanin and Rolnick [1,2] on the expressivity of PieceWise Linear (PWL) Deep Neural Networks (DNNs).
The main Contribution (**C1**) involves improving the theoretical bounds of both the number of linear regions and the average distance to the linear boundaries defined by a PWL DNN, by taking into account the geometry of the input manifold. The authors also provided a toy example to empirically validate the results. In addition, they performed an experiment of natural images (**C2**), to empirically show that both memorization and closeness to the data manifold have an effect on the density of linear regions.

### References
- [1] Boris Hanin and David Rolnick, “Deep ReLU Networks Have Surprisingly Few Activation Patterns”, NeurIPS 2019.
- [2] Boris Hanin and David Rolnick, “Complexity of Linear Regions in Deep Networks,” ICLR 2019.

**Questions:**

### Questions
- With regard to (**W1**), to my understanding, in [1] (Theorem 1, page 3) the authors claimed that "The same result holds when computing the number of linear regions along any fixed 1-dimensional curve in a high-dimensional input space.", confirming the results obtained in Figures 6 and 7. The informal (not proven) claim in [1] about the exponential growth of regions with the input dimension in my opinion refers to the input manifold size $m$. In this regard, it would have been interesting to test the number of regions with increasing size $m$ of the manifold. Please better clarify the statement you are referring to and why your empirical results are in contrast with it.
- With regard to (**W2**), I didn't fully why should you claim in Sect. 4.3 that overfitting is the cause of the decrease in the number of regions during training. In my opinion, it could be a result of the chosen hyperparameters. Can you clarify your intuition?

### Minors and suggestions
- section 4.1 could have been more clear. In particular, please explicit that the domains of Sphere and Tractrix are similar for the sake of comparability, i.e. $[-\pi, \pi]$ and $[-3, 3]$, and that the parameters of the target function are the same.
- for the sake of consistency, are the x, y and z-scales in Figure 1 (a) the same as the ones used in the experiments?
- please explain how you computed the density in Figure 9.
- please report the performance of the PWL DNN used in Figure 9.
- use either “piece-wise” or “piecewise” in the text.
- line 248 (main): Figure 11 is probably Figure 2 of the Appendix.
- line 264: in the equation of the tractrix y should be t.
- line 22 (Appendix): typo of the subscript.
- broken refs in the Appendix.

### References
- [1] Boris Hanin and David Rolnick, “Complexity of Linear Regions in Deep Networks,” ICLR 2019.


**Limitations:**

I would have addressed the role of hyperparameter selection in the experimental results since previous work clearly stated that size, learning rate, and other hyperparameter choices have an effect on the evolution of the number of linear regions during training.

**Strengths And Weaknesses:**

### Strengths
- (**Strength 1**) The paper addresses the significant problem of extending the bounds on expressiveness metrics of PWL DNNs in [1,2], by taking into account the dimensionality and geometry of the input manifold.
- (**S2**) I really liked the toy example as a way to validate the analytical results, although the clarity could be improved.
- (**S3**) The authors helped the reader in understanding the intuition behind the theoretical results by providing useful Figures (1 and 2 in the main, 1 and 2 in the Appendix).

### Weaknesses
With regard to the weaknesses, some of the experimental results (2 out of 3) are not sufficiently clear or do not provide a significant contribution:
- (**Weakness 1**) **Sect. 4.2**: The authors claimed to have obtained results (Figures 6 and 7) in contrast with [2]. However, it is not clear to me which result in [2] the authors are referring to (see Questions).
- (**W2**) **Sect. 4.3**: the MetFaces experiment represents an additional, separate contribution (**C2**), but due to lack of space the results were not validated by enough empirical tests. For example, claiming a relationship between the decreasing number of linear regions during training and memorization was not validated against any hyperparameter change, as it was done in [1]. In fact, the behaviour of the number of regions in an overfitting scenario was shown in [1] to be dependent on both the number of points to be memorized against the capacity of the network and the performance of the model (not reported).
- (**W3**) **Sect. 4.3**: the second result of the MetFaces experiment, i.e. the difference in the number of regions in and out of the data manifold was already shown in [3], Figure 2, with a slightly different experimental setting. Although using a GAN to fit the in-distribution manifold is a nice idea, the obtained result, in my opinion, doesn't contribute to the overall significance of the paper.

### Concluding Remarks
Although the paper addresses a significant research question, validated with a toy example, I think that more empirical experiments should have been devoted to the main contribution (the role of the manifold geometry), instead of focusing on the role of memorization and closeness to the data manifold (both already analyzed in-depth in previous papers [1,3]). Moreover, I have doubts about some of the claims resulting from the experimental results, which either are not clear or do not address the limitation of the experimental evaluation.

### Reviewer-Authors Discussion
After the reviewer-authors discussion, I increased the rating of both soundness and presentation. Moreover, I increased the overall rating from 4 to 6. Please refer to the discussion for the details.

### References
- [1] Boris Hanin and David Rolnick, “Deep ReLU Networks Have Surprisingly Few Activation Patterns”, NeurIPS 2019.
- [2] Boris Hanin and David Rolnick, “Complexity of Linear Regions in Deep Networks,” ICLR 2019.
- [3] Roman Novak et al., “Sensitivity and Generalization in Neural Networks: An Empirical Study.” ICLR 2018.

---

> ### Author Response · Authors · 2022-08-02
> **Response to Reviewer J3BE**
>
> Thank you for the useful citations, careful consideration of our work, and very useful feedback.
>
> On weakness 1: With Figures 6 and 7 we point out how the dependence of the “effective” number of linear regions is not exponential in the input dimension but depends instead on the intrinsic dimension. This furthers our argument for studying the density of boundary regions on the data manifold and not the ambient space. We will clarify this in our work and not phrase it as “stand in contrast”. We have rephrased it in section 4.2. Does this sufficiently address your concern?
>
> On W2: Thanks to you pointing out how past work has already studied overfitting, we have now dropped the claim that the density of linear regions decreases in case of overfitting after further experimentation. We ran the same experiment for additional hyperparameters and found that the pattern is not reproducible. We report these results in Appendix I.
>
> On W3: We thank you for pointing us to [3] since we had not included it in the related works section and have rectified it in the newest version. While [3] does study the transition boundaries on and off the manifold we present a novel empirical perspective on high-dimensional data with respect to the density of linear regions and the behavior under pathological case of overfitting. To the best of our understanding,  [3] studies transitions in the input space using linear combinations and instead we generate curves on the manifold of MetFaces using GANs to study this density. This is both a novel approach and a novel empirical observation in a high dimensional case. Please let us know if this is incorrect.
>
> Response to the questions:
>
> Q1: We thank the reviewer for pointing us to the relevant line in [1]. We note that while their result holds for the 1-D curve in higher dimensions ours has an added term for the geometry of this curve (or the manifold itself). While we agree it would be interesting to observe the increase in linear regions as m increases, we have still communicated the central message of our work using the experiments: the “effective” density of linear regions depends on the dimensionality of the data manifold and not the input dimension. We show this empirically for m=1. As of now, we do not have a tractable method for calculating the density of the number of linear regions in higher dimensional manifolds.
>
> Q2: Please see our response to W2 above.
>
> We address some of your minor suggestions:
> 1. We have added further information in the Appendix H about the experiments and we have pointed to it in the main body. Also, we note how the domains are the same in Section 4.1.
> 2. Could you please tell us what is Figure 1(a) here? Are you referring to the Appendix?
> 3. We have added further details in both Appendices H and I on computing these densities let us know if this explains the procedure sufficiently.
> 4. We have added the training and test accuracy graphs for various hyper parameters in Appendix I (Figure 5).

---

> > ### Comment · Reviewer_J3BE · 2022-08-04
> > **Rebuttal comments**
> >
> > First of all, I would like to thank the authors for having carefully taken into account all my considerations and doubts.
> >
> > I consider my concerns about (W1), (W2) and (W3) as addressed. Concerning the MetFaces experiment (W3), I think that with some clarity improvements it will now fit better the scope of the paper.
> >
> > To recap after the last version, the main contribution of the paper is to highlight the impact of the input data geometry on the expressivity of DNNs. This was shown by providing theoretical bounds on an expressivity proxy (density of the boundaries of linear regions). Regarding the soundness, the theoretical bounds were validated with a toy example. Moreover, additional experiments show that when evaluating the required expressivity for a DNN to learn a given task, one should consider:
> > 1. the intrinsic and not the extrinsic dimensionality of the input.
> > 2. the in-distribution manifold only, since the out-distribution is expected to have a smaller linear region density.
> >
> > As I said, the overall clarity and motivation of the experiments should be improved to highlight their contribution. As an example, the abstract still reports "We further demonstrate how the complexity of linear regions changes on the low dimensional manifold of images as training progresses, using the MetFaces dataset.", but I think that (after the last changes) the analysis is more about the in- vs out- of the manifold density, rather than the evolution of the density during training.
> >
> > About the minors, I was referring to Figure 3a of the main, which should (in my opinion) either have the same scale (domain) of the experiments or no axis ticks at all, as to not confuse the reader.
> >
> > All in all, I think that after improving the clarity of the experimental results and their contribution, I will be more toward acceptance.

---

> > > ### Author Response · Authors · 2022-08-06
> > > **Added experimentation details**
> > >
> > > Thank you for your continued engagement and useful feedback on our work. We hope that our latest revision answers some of the concerns.
> > >
> > > We agree that the last changes have narrowed our focus to the complexity of linear regions on the manifold vs the Euclidean space. We have changed the introduction, discussion section and abstract to reflect this. In addition to that we have also added further experimental details. To summarise the changes on the experiment side:
> > > 1. We have added a paragraph at the beginning of Section 4, in the main body, on how we count the number of linear regions by solving an equation in one variable. This method is used for all the experiments.
> > > 2.   We have also added details on how we construct the curves for the MetFaces dataset on vs off the manifold.
> > >
> > > We have tried to incorporate all the changes with space constraint in mind. Please do let us know if this answers all the concerns?
> > >
> > > On Figure 3(a): We have added a note that this is only for illustration purposes and not the exact same domain or target function. The reason being that with a smaller domain the image does not illustrate the geometry of the tractrix very well.

---

> > > > ### Comment · Reviewer_J3BE · 2022-08-08
> > > > **Rating increase**
> > > >
> > > > I would like to thank again the authors for their work and consideration of my observations. I think that with the last changes the presentation quality has definitely improved. As a result of both the rebuttal and the discussion, I raised the overall rating from 4 to 6.

---

### Official Review · Reviewer_QPts · 2022-07-11

**Rating:** 6
**Confidence:** 2
**Soundness:** 3 good
**Presentation:** 3 good
**Contribution:** 3 good

**Summary:**

This paper studies the data geometry of randomly initialized deep neural networks with ReLU activation. Motivated by the manifold hypothesis, the authors present an extension of the previous result in Hanin and Ronick [1], where the previous assumption of uniform sampled data on the entire input space is relaxed to sampling on submanifolds. The authors develop analogous upper bounds on the density of boundary regions, and lower bound on expected distance to the boundary. The authors then verified their results empirically using both synthetic datasets and the MetFaces dataset.

[1] B.Hanin and D. Ronick. Complexity of linear regions in neural networks, ICML 2019.

**Questions:**

I am looking at your final experiment on the MetFaces dataset (in Figure 9).
- While I do agree that the neural network would eventually overfit, I would like to understand when the network starts to overfit, and whether that corresponds to the epoch when the number of linear regions starts to “plateau”.
- Have you tried comparing the plots of number of linear regions, together with the evolution of training/testing error of the network? Are there any correlation in the trends of these plots?

**Strengths And Weaknesses:**

__Quality and Clarity__: The paper is in general well-written. I enjoyed reading the motivations of the paper, as well the interpretation sections for the theorems. While I have not checked the proofs carefully, the progression of the theorems make sense to me. The experiments also empirically support the author’s interpretations, though I have some questions which I will ask in the next section.

__Originality and Significance__: The paper is mainly an extension of the previous Hanin and Ronick paper onto the submanifold case. While the framework here is not new, I believe it is still a novel result, as the relaxed assumption of the data lying on a submanifold is indeed more realistic. That said, I am personally not a researcher in this field, and I would refer to other reviewers for further evaluation of the significance of the results.

---

> ### Author Response · Authors · 2022-08-05
> **Response to Reviewer QPts**
>
> We thank you for your insightful comments. We answer your questions below.
>
> *when the network starts to overfit, and whether that corresponds to the epoch when the number of linear regions starts to “plateau”.*
>
> We have now included the graphs for the accuracy in Appendix I, Figure 5 (over multiple hyperparameters). Surprisingly, the accuracy and the number of linear regions do not plateau at the same epoch. This is not true for the toy problem (comparing Figure 3 in Appendix with Figure 4 in the main body of the revised version). Therefore, we cannot make any assertions as of now on this topic. This has also been seen in the past (Figure 4 in [1]).
>
> *Have you tried comparing the plots of number of linear regions, together with the evolution of training/testing error of the network? Are there any correlation in the trends of these plots?*
>
> This has been studied in the past as well although not on manifolds [1, 2]. From the graphs for the case with generalization (the toy problem), we can assert the same as previous studies that the number of linear regions increases as the accuracy increases and then plateaus accordingly, albeit at different rates. We will add this observation, which follows past empirical results, to the revised version of our work.
>
> **References:**
>
> [1] Boris Hanin and David Rolnick, “Deep ReLU Networks Have Surprisingly Few Activation Patterns”, NeurIPS 2019.
>
> [2] B.Hanin and D. Ronick. Complexity of linear regions in neural networks, ICML 2019.

---

### Official Review · Reviewer_o5b5 · 2022-07-12

**Rating:** 7
**Confidence:** 1
**Soundness:** 4 excellent
**Presentation:** 4 excellent
**Contribution:** 3 good

**Summary:**

With the assumption that a dataset is sampled from a low dimensional manifold, this paper incorporated such data geometry into measuring the effective approximation capacity of DNNs by deriving average bounds on the density of boundary regions and distance from the linear boundary, which is reported in previous work by Hannin and Rolnick.

**Questions:**

Would it be possible to provide an example of how the findings of this study can help with users of deep neural networks to design, explain, and improve their systems?

**Limitations:**

As mentioned by the authors, proving a lower bound on the number of linear regions still remains open.

**Strengths And Weaknesses:**

Strengths:

Originality: this work builds upon previous work by Hannin and Rolnick, which establishes the linear regions in deep networks, and provides an original study on how neural networks with ReLU activation can split data manifolds into regions where the neural network behaves as a linear function.

Quality: the study is of relatively good quality. Bound on the number of linear regions and the distance to boundaries of these linear regions on the data manifold are derived.

Clarity: the paper is very clearly presented.

Weaknesses:

Significance: while building upon Hannin and Rolnick's previous work, the significance of this extension is not very clear, especially how this work leads to understanding the expressivity of deep networks on non-Euclidean data sets.

---

> ### Author Response · Authors · 2022-08-04
> **Response to Reviewer o5b5**
>
> We thank you for engaging with our paper and asking insightful questions for further clarifications. We understand that the primary confusion is with respect to how this increases the understanding of the expressivity of the DNNs and also how it can lead to better practice.
>
> The main message of our work is that we ought to be studying the expressivity of DNNs on the data manifold and not on the entire Euclidean space. This follows a line of existing work in deep learning theory [1, 2, 3], which is covered in the related works section in Appendix C. We show that there is a distinction between bounds derived by Hanin and Rolnick, 2019 for the density of boundary regions on a compact set in the Euclidean space as compared to our bound for density on the Manifold. We demonstrate empirically how this difference in bounds manifests itself using a toy problem.
>
> On  "how the findings of this study can help with users of deep neural networks to design, explain, and improve their systems?'":
>
> We provide one experiment using the high dimensional MetFaces dataset and show how these densities are different on and off the manifold. We provide one important insight for practitioners:
>
> *The density of linear regions is significantly lower on the data manifold and devising methods to “concentrate” these linear regions309
> on the manifold is a promising research direction.*
>
> To provide an example on how to implement this insight would be beyond the scope of our current work. We still argue that this opens up avenues for future research.
>
> References:
> [1] Uri Shaham, Alexander Cloninger and Ronald R. Coifman, Provable approximation properties for deep neural networks, 2015.
>
> [2] Alexander Cloninger and Timo Klock, ReLU nets adapt to intrinsic dimensionality beyond the target domain, 2020.
>
> [3] Minshuo Chen, Haoming Jiang and Wenjing Liao and Tuo ZhaoEfficient Approximation of Deep ReLU Networks for Functions on Low Dimensional Manifolds, 2019.

---

### Official Review · Reviewer_gb2e · 2022-07-13

**Rating:** 5
**Confidence:** 3
**Soundness:** 3 good
**Presentation:** 2 fair
**Contribution:** 2 fair

**Summary:**

The paper studies how randomly initialized ReLU networks split the data into linear regions. It is based on Hanin and Rolnick [2019a] but focuses on data with low dimensional manifold structure.  It provides an upper bound for the density of the linear regions, which results in a lower bound for the average distance to the boundary of all linear regions. This lower bound is inverse linear in the number of neurons and related to the geometric properties of the manifold. The authors then empirically verify their statement over both synthetic datasets of two parameterized curves and one realistic dataset of a curve generated from StyGAN.

**Questions:**

The questions would be related to some of the comments above.

1. The role of dimensionality and curvature. Are we able to relate $C_M$ and $C_{M, \kappa}$ to some intrinsic characterization of the manifold, like dimension or scalar curvature? How will the dimensionality of the manifold affect the number of linear regions or the average distance? Also, $C_{M, \kappa}$ is defined by $p_U(\zeta)$, which has an inverse dependence on the number of neurons. Why $C_{M, \kappa}$ will not depend on the number of neurons?

2. The role of the depth of the network. One key observation in Hanin and this paper is that the number of linear regions has minimum dependence on the number of neurons. On the other hand, a deeper network would have more expressive power. It would be nice to observe this through experiments. In figure 8, there are attempts to understand how different choices of the network structure affect the number of linear regions. However, they are all two-layer networks. If one tries a deep network structure, will the result remain the same?

3. As argued in section 3, one cannot apply Hanin's result directly because $Vol_{n_{in}}(M)$ is $0$. One can consider an $\epsilon$ neighborhood of $M$ and try to bound the volume by the geometric properties of the manifold. How is this approach different from the one in the paper? In other words, will this approach give non-optimal bounds compared to looking closer to the intrinsic characterization of the manifold?


**Limitations:**

The authors have adequately addressed the limitations and potential negative social impact of their work.

**Strengths And Weaknesses:**

Strength:
The problem this paper trying to solve is interesting. In many applications, data exhibit low dimensional or low dimensional manifold structures. Understanding how the intrinsic dimension and geometries interact with the deep network is an interesting problem. The paper is able to relate the average distance from the linear region to parameters depending on the scalar curvature and dimensionality of the manifold, while still preserving the same dependence on the number of neurons as in the Euclidean case. The experiments are insightful and support the intuitions of the theory.

Weakness:

1. The paper mostly serves as a theoretical paper, but some key definitions are not presented in a clear and rigorous way. The paper is partially based on Hanin 2019 so the authors seem to adopt some of the notation in that paper without further clarification. For example, the definition of $\mathcal B_{F, k}$ defined in line 162 is not clear in the first read. I need to go and forth between this and Hanin's paper to understand a similar definition and the relationship between them. I would encourage the author to provide a formal math definition of $\mathcal B_{F, k}$ in the appendix.

2. The role of the geometries of the manifold is not clearly laid out in the theorem. The key benefit of considering data with structure is to understand how the structure interacts with the neural networks. Theorem 3.4, which is the main theorem of the paper, is very similar to the Euclidean case in the sense that it has inverse dependence over the number of neurons. However, the definition for $C_M$ and $C_{M, \kappa}$ is not provided in an intuitive way for the readers to understand the relationship between the geometries of the manifold and average distance. For example, one would like to know how $C_M$ and $C_{M, \kappa}$ depend on the scalar curvature and dimension of the manifold. In the current form, it would be hard for the reader to understand the exact benefit of considering data with low dimensional manifold structure.

3. The linear region density (left-hand side of equation on line 145 or line 202) only serves as a proxy for the number of linear regions. In the later work of Hanin and Rolnick [2019b], the number of linear regions would still depend on $(\text{neurons})^{n_{in}}$. So Theorem 3.3 does not provide a bound for the number of linear regions directly.


Minors:
1. Line 126, it should be $z(x) = W_{l-1}\sigma(\cdots \sigma(B_1 + W_{1}(x)))$. In line 127, it should be $z(x) = W_{1, z}(x)$.
2. Line 22 in the appendix should be $g_{ij}(x)$.
3. Many reference links in the appendix are broken.
4. In appendix line 216, the definition of $Ker(D_MH_k)^\perp$ should be $Ker(D_MH_k)^\perp = \set{a \vert a \cdot z = 0 \forall z \in Ker(D_MH_k)}$ instead of for some $z$.

---

> ### Author Response · Authors · 2022-08-02
> **Response to Reviewer gb2e**
>
> Thank you for the thorough consideration of our paper and the valuable feedback. We address your concerns one after another below.
>
> 1. On the definition of the boundary regions $\mathcal B_{F, k}$ - thank you for pointing out that this was hard to parse. We note that this set is defined in lines 161-164, which is based on the definition of $\mathcal B_{F}$ below line 135. We agree that we can elaborate on the definition of $\mathcal B_{F, k}$. We have done so in Appendix E. Please let us know if any other definition or elaboration is required.
>
> 2. We agree that the average distance depends inversely on the number of neurons (Theorem 3.4). We have tried to provide sufficient explanation for definitions of $C_M$ and $C_{M, \kappa}$, and would like to point you to Figure 2 in the Appendix where we explain how $C_M$ captures how vectors shrink differently on different manifolds. We are unsure of how curvature affects this quantity and that is beyond the scope of our current work. Having said that, solely based on the definition we would argue that this is a quantity intrinsic to the manifold. We also provide an exact definition for $C_{M, \kappa}$ which depends on a polynomial of the curvature at various dimensional sub-manifolds. We also explain how curvature changes the volume of a geodesic ball on the manifold, intuitively, in Figure 1 of the Appendix whilst providing the exact definition of the scalar curvature being used. Since these are theoretical constants depending on the geometry of the data manifold it is difficult for us to provide further intuition. Despite this we have provided intuition on these constants in the appendix. We will ensure that the pointers to these intuitions are added to section 3.1. We have added the following information for $C_{M, \kappa}$ in section 3.1:
> “The constant $C_{M, \kappa}$ depends on the curvature polynomially, with order at most $n_{in}$.”
> On “it would be hard for the reader to understand the exact benefit of considering data with low dimensional manifold structure” -> we make it very clear how the density of linear regions has this dependence on the dimensionality of the data manifold and not the ambient space and this has an exact bearing on the “relevant” density of boundary regions. We also show that there are constants governing this density. While we do not show exactly how these constants affect which DNN architecture is a better fit, we have provided some intuition based on these constants. Getting to an exact benefit will be subject to future work and is beyond the current scope but we believe strongly that this provides a meaningful direction in that regard. In the theory of DNNs, past work has generated promising research directions and our work follows suit in that sense.
>
> 3. On the fact that we are not providing bounds on the density of the number of linear regions. For the 1 dimensional manifold theorem 3.3 is a bound on the density of linear regions (since 0-d volume of boundary regions is a count of the number of boundaries). For the general case and higher dimensions, we do agree that this is a bound on the density of boundary regions which serves as a proxy for the number of linear regions. We have changed the abstract to reflect that. However, we do acknowledge how this density of boundary regions is a proxy measure on lines 31 and 138 in the main body of the paper.
>
> Answering the questions raised:
> 1. On “how the constants depend on the intrinsic characterization of the manifold”: we have addressed some of the concerns in response to point 2 above. To elaborate further on the constant $C_{M, \kappa}$, it is a supremum of all possible polynomials depending on the number of neurons. The value $\xi$ can only be changed by varying the number of neurons and thereby eliminates the dependence of the constant on the number of neurons. We then use a much simpler dependence on the inverse of number neurons to explain how it depends on the architecture. To be more precise, $C_{M, \kappa}$ is defined by a supremum on $p_U(\xi)$ over $\xi$ and not the polynomial $p_U(\xi))$ for a fixed $\xi$. This is akin to finding an extrema within a limited domain of a polynomial. Note that exponents of this polynomial are dependent on the manifold dimension and the input dimension.
>
> 2. While we do agree the depth would change the expressive power of the DNN we also think that understanding this is beyond the current scope of our work. Note that Hanin and Rolnick, 2019 provide additional experiments for this variation in their work. Our primary objective is to demonstrate the dependence of the data geometry under fixed architecture since past work has dealt in great detail with the dependence of depth.
>
> Continued ...

---

> > ### Author Response · Authors · 2022-08-02
> > **Response to Reviewer gb2e continued**
> >
> > 3. We thank you for pointing this novel approach. We are unsure if the new approach of considering an $\epsilon$ neighborhood of $M$ is better or worse. The argument itself would be much trickier since we would be now calculating an $\epsilon$-neighborhood proxy for density of linear regions which is off the data manifold and moreover would require very careful consideration for what happens when these neighborhoods overlap. The geometry of the manifold will play a role but we believe our approach is a more direct way of analyzing the density of boundary regions.
> >
> > We have also fixed the minor issues pointed out by you in the latest version. We have also added train and test accuracy graphs for various hyperparameters in the Appendix.

---

> > > ### Comment · Reviewer_gb2e · 2022-08-08
> > > **Thanks But Further Concerns Over the Dependence of $C_{M, \kappa}$ Over Input Dimension**
> > >
> > > I thank the authors for their detailed and helpful responses to my comments. I believe most of my concerns are well addressed for now.
> > >
> > > For comment 1, I consider lines 161-164 not a formal definition and hard to parse. The additional definitions in appendix E serve the purpose well.
> > >
> > > For comment 2, I now agree with the authors that the authors provide enough intuitions and explanations for the definitions of $C_{M}$ and $C_{M, \kappa}$, so this part of the concern is well addressed.
> > >
> > > In my earlier comments, what confuses me is the unclear relationship between $C_M, C_{M, \kappa}$ and the intrinsic properties of the data. As authors pointed out, the primary objective is to show that these quantities only depend on intrinsic properties of the data including intrinsic dimensions and curvatures, rather than extrinsic properties like input dimension. Still, one wants to understand how can we can compare the result with the previous one obtained in Hanin and Rolnick [2019a]. In other words, I would like to know the benefit I can get when data has a low-dimensional structure. I am not expecting a closed form formula for $C_M$ and $C_{M, \kappa}$ as the dependencies over geometries are complicated, but am looking for appropriate bounds or even examples to help me understand how large they can be.
> > >
> > > In the responses, the authors have added comments in section 3.1 that "The constant $C_{M, \kappa}$ depends on the curvature polynomially, with order at most $n_{\text{in}}$." This is actually the type of results I am looking for. Is this result stated somewhere formally in the appendix? On the other hand, it actually created further confusion for me. In the statement of theorem 3.4, it is claimed that "$C_{M, \kappa}$ depends on the scalar curvature and dimensionality of the manifold $M$". I will not expect a dependence over $n_{in}$ from such a statement. How should I understand this? To further clarify my confusion, I would like to know how the results in the paper depend on the input dimension $n_{\text{in}}$ and manifold dimension $m$, and how should I compare it with the results for the Euclidean case. The dimensionality here is a quantity more important than curvatures, so I can accept a terrible dependence over curvature but hope the dependence over $n_{\text{in}}$ and $m$ can be further clarified.

---

> > > > ### Author Response · Authors · 2022-08-08
> > > > **The dependence on $C_{M, \kappa}$ over input dimension**
> > > >
> > > > Thank you for engaging with our rebuttal and your continued patience in the review of our work.
> > > >
> > > > Your clarifying comment on *how the constant $C_{M, \kappa}$ depends on the ambient dimension?* helped us think further about this issue and we provide more information on how it varies with $n_{in}$ and $m$ in the latest revision. Since  $C_{M, \kappa}$  is a supremum of a polynomial whose exponents depend both on $n_{in}$ and $m$, we took the simple example of an analogous polynomial $p_{\text{simplified}}(\zeta) = \zeta (1 - \sum_{k = n_{in} -m}^{n_{in}} \zeta^{k})$ and plot how the supremum varies, for the restricted domain $(0, 1)$, with $n_{in}$ with $m$ fixed and $m$ with $n_{in}$ fixed. We provide these results in Appendix G.1. See plots in figures 3 and 4 in the Appendix as well. Please note that we do this for a simplified polynomial because otherwise it would be very difficult to observe the changes in the supremum with changes in the coefficients. We have also refined the language in lines 248-252 of the main body of the paper describing the effect of $n_{in}$ and $m$ on the constant $C_{M, \kappa}$, and in Theorem 3.4 with a caveat that the constant depends on $n_{in}$.
> > > >
> > > > Thank you once again for your insightful comments that have helped us improve our work. We hope that this addresses the above concern.

---

> > > > > ### Comment · Reviewer_gb2e · 2022-08-09
> > > > > **Further Comments**
> > > > >
> > > > > I thank the authors for their feedback and further illustrations and experiments. The additional experiments seem to verify that $C_{M, \kappa}$ is an increasing function over $n_{\text{in}}$ and decreasing function over $m$. Even though it is not clear that it will be of scale $\text{curvature}^{n_{\text{in}} - m}$ as figure 4 in the appendix does not seem to be exponential in $n_{\text{in}}$.
> > > > >
> > > > > After all the discussion and reevaluation of the work, I am willing to increase my rating from 4 to 5. However, I cannot further increase my score because I am still confused about the dependence of $C_{M, \kappa}$ over input and manifold dimension, and the implications of that. I would like to apologize for the late responses in the discussion period that failed to provide the authors enough time for further explanation. If the paper is accepted, I would recommend the following changes to clarify the problem.
> > > > >
> > > > > 1. Instead of nice experiments in figure 3 and 4 in the appendix, provide toy examples where the dependences over $n_{\text{in}}$ and $m$ can be computed/ bounded in closed form. The authors still claim in section 3.1 that "The constant $C_{M, \kappa}$ depends on the curvature as the supremum of a polynomial whose coefficients depend on the curvature, with order at most $n_{\text{in}}$ and at least $n_{\text{in}} - m$." This is more a formal statement than an intuitive guessing. It would be nice if the authors can convince readers about this point.
> > > > >
> > > > > 2. Related to point 1, I consider examples with simple geometries would serve the purpose well. For example, the authors can provide the exact formula for
> > > > > 1) data that lies on a $m$ dimensional linear subspace
> > > > > 2) data on a $m$ dimensional sphere.
> > > > > This would help readers to interpret the role of the curvature and intrinsic dimension super clearly. Furthermore, one can use these examples and their results to compare with the Euclidean space results in Hanin and Rolnick, 2019a, b, to argue about the benefit of considering a low dimensional structure.

---

> > > > > > ### Author Response · Authors · 2022-08-09
> > > > > > **Response to further comments**
> > > > > >
> > > > > > We thank you for your continued interactions. We respond to your comments here.
> > > > > >
> > > > > > Thank you for going over our additions to the Appendix in explaining how a simple polynomial's optima changes and eventually effects the constant $C_{M, \kappa}$. We have tried our best to provide intuition for various constants we do agree we can add more intuitive explanations which we plan to do in future versions.
> > > > > >
> > > > > > On point 2 above:
> > > > > >
> > > > > > We have compared the average distances for two different geometries in the experimentation section and shown how this affects the quantity. This in turn gives us a distinction between how both the geometric constants in Theorem 3.2 vary across geometries. This does validate our theorem. We do not have an exact formula for these constants even in the case of simple geometries such as the circle and tractrix. Having said that, our main goal was to show the differences in the density of linear boundary regions on different manifolds and we have shown just that. We can also do so for the linear 1D case and contrast it with Hanin and Rolnick, 2019 in the future version (since we are almost out of time here), which will serve as an addition to our argument. Another unfortunate aspect of presenting experiments with varying $m$ is that counting the number of linear regions efficiently for large networks on a 2D plane still remains an open problem and has not been done successfully in the past [1,2,3] all the past studies do so for 1D curves, which makes doing so for higher dimensions of $m$ much more complicated.
> > > > > >
> > > > > > **References:**
> > > > > > [1] Boris Hanin and David Rolnick, “Deep ReLU Networks Have Surprisingly Few Activation Patterns”, NeurIPS 2019.
> > > > > >
> > > > > > [2] Boris Hanin and David Rolnick, “Complexity of Linear Regions in Deep Networks,” ICLR 2019.
> > > > > >
> > > > > > [3] Roman Novak et al., “Sensitivity and Generalization in Neural Networks: An Empirical Study.” ICLR 2018.

---

### Meta-Review · Area_Chair_MsQ9 · 2022-08-30

**Recommendation:** Accept
**Confidence:** Certain

**Metareview:**

The paper studies the number of linear regions cut out by a randomly initialized deep network, for data with low-dimensional structure (manifold structured data). The main results pertain to the density of linear regions and the average distance to the boundary of a linear region: these results take the same form as in the Euclidean case (with distance inversely proportional to the number of neurons), but depend on geometric properties of the data manifold — in particular, its dimension and curvature. Reviewers generally appreciated the relevance of the paper’s setting: data arising in applications often have low-dimensional structure, and understanding how deep networks interact with the structure of data is an important research direction. At a technical level, the paper builds on techniques of [Hanin and Ronik 2019], but extends these results to manifold structured data. Questions raised by the reviewers include the role of curvature and input dimension in the results and the interpretation of real data experiments. After interacting with the authors, the reviewers considered their main concerns about the paper to be well-addressed. The AC concurs, and recommends acceptance.

**Award:**

No

---

### Decision · Program_Chairs · 2022-09-14

Accept